# The Effect of Quercetin on the Growth, Development, Nutrition Utilization, and Detoxification Enzymes in *Hyphantria cunea* Drury (Lepidoptera: Arctiidae)

Yi-Lin Gao [1],[†], Zhong-Yu Pan [1],[†], Xiang Meng [1],[†], Yu-Fei Yuan [1], Hong-Yun Li [2] and Min Chen [1],*

[1] Key Laboratory of Beijing for the Control of Forest Pests, Beijing Forestry University, Beijing 100083, China
[2] Management Office of the Temple of Heaven Park, Beijing 100061, China
* Correspondence: minch@bjfu.edu.cn; Tel.: +86-18611154462
[†] These authors contributed equally to this work.

**Abstract:** *Hyphantria cunea* Drury (Lepidoptera: Arctiidae) is a worldwide quarantine pest that has a wide range of host plants. Quercetin is a secondary metabolite involved in chemical defense processes in plants. To understand how *H. cunea* adapt to quercetin in its host plants, we determined the effects of quercetin on larval mortality, growth, nutritional indices, and the activity or content of detoxification enzymes in *H. cunea* larvae by feeding them an artificial diet containing different concentrations of quercetin. Our results showed that 0.50% quercetin treatment significantly prolonged the development duration of *H. cunea* larvae and inhibited growth of *H. cunea*. Nutritional indices analysis indicated that quercetin significantly affected nutrient use, including effects on the approximate digestibility, consumption index, relative growth rate, and efficiency of conversion of ingested food to body substance. Furthermore, our results revealed that quercetin reduced the content of carboxylesterases, and increased the activity or content of glutathione S-transferases, UDP-glucuronosyltransferases, and ATP-binding cassette transporters in *H. cunea* larvae. These results provide a foundation for revealing the adaptation that *H. cunea* use to adapt to quercetin in host plants.

**Keywords:** *Hyphantria cunea*; quercetin; mortality; nutrient utilization; detoxification enzymes

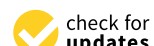



## 1. Introduction

Plants have developed various defense systems to combat insect attacks during the prolonged interactions and coevolution between herbivorous insects and plants [1]. Plant secondary metabolites play a defensive role in the resistance of plants to various phytophagous insects [2,3]. The effects of secondary metabolites on insects include strategies that prohibit feeding, repelling and poisoning insects, and mechanisms that inhibit insect growth and development [1,4,5]. In addition, secondary metabolites can attract predators of herbivorous insects, forming an indirect defense pattern [6].

The success of phytophagous insects is dependent on adapting to the changing biotic stress of different secondary metabolites in the host plant and accordingly modulating their defense states [7]. To achieve this, phytophagous insects have developed sophisticated behavioral and physiological defense mechanisms [8]. Diet stress from exposure to plant chemicals can cause changes in herbivore physiology and gene expression [9]. Insects consume a lot of energy to detoxify plant secondary metabolites, which is achieved by adjusting nutritional strategies and through energy redistribution [10]. In addition, detoxification enzymes play a crucial role in the response of insects to plant secondary metabolites [11]. Through the action of detoxification enzymes, insects modify ingested chemical toxins and render them less toxic, by suppressing gut enzymes, and easier to transport or excrete [12]. The main insect detoxification enzymes include cytochrome P450 monooxygenases (CYP450), glutathione S-transferases (GSTs), carboxylesterases (CarE),

UDP-glycosyltransferases (UGTs), and ATP-binding cassette (ABC) transporters [13,14]. Adjustments in the nutrition utilization strategy and changes in detoxification enzymes are the key adaptations used by insects to adapt to host plants [15,16].

The fall webworm, *Hyphantria cunea* Drury (Lepidoptera: Arctiidae), is a quarantine pest worldwide. The insect originated in North America, spread to Asia, Europe, and the Americas, and is now present in about 35 countries [17]. In China, *H. cunea* has spread to 608 county-level administrative districts of 14 provinces since its first occurrence in Dandong City of Liaoning Province in 1979 (China National Forestry and Grassland Administration Announcement No.7, Dandong, China, Luo 2021). *H. cunea* is a polyphagous pest with many host plants, and is estimated to globally forage more than 600 plant species [18,19]. The robust adaptability of *H. cunea* to multifarious secondary metabolites in hosts is one likely reason for its rapid spread [20]. However, the polyphagous survival strategy of *H. cunea* and the detoxification mechanisms used to combat plant secondary metabolites have rarely been studied. Wang [21] found that *H. cunea* resisted the adverse effects of plant secondary metabolites through an antioxidant system, detoxification mechanism, and by increasing its food consumption rate.

Quercetin is one of the most ubiquitous and abundant flavonoids in the plant kingdom and its role is assumed to protect plants against herbivores and pathogens [22–24]. Quercetin is contained in hosts plants, such as *Morus alba* L. [25] and *Platanus orientalis* L. [26], and non-hosts plants, such as *Ginkgo biloba* L. [27]. Quercetin inhibits the growth of *Helicoverpa armigera* Hb. (Lepidoptera: Noctuidae) and *Spodoptera exigua* Hübner (Lepidoptera: Noctuidae) [28,29], and reduces the pupation rate of *H. armigera* and *S. litura* [30,31]. From the perspective of mortality, quercetin increases the larval mortality of *S. frugiperda* (Lepidoptera: Noctuidae), *Oedaleus asiaticus* Bey-Bienko (Orthoptera: Acrididae), and *Lymantria dispar* L. (Lepidoptera: Lymantriidae) [32–34]. Further, quercetin changes the activity of detoxification enzymes and alters insect gene expression levels [35,36]. It also enhances the actions of CYP450, GSTs, and CarE in the midgut and fat bodies of *H. armigera* and *S. litura* [37–39], in addition to inhibiting the activity of CarE in *L. dispar* and GSTs in *Closter anachoreta* (Lepidoptera: Notodontidae) [34,40]. However, the effect of quercetin on the growth, development, and physiological adaptive response of *H. cunea* are unclear.

Due to the limited information available on the adaptation mechanisms of *H. cunea* in a wide range of host plants, the present study explores the effects of quercetin on the growth and development, mortality, nutritional indices, and detoxification enzyme activities or contents of *H. cunea* larvae. With the addition of quercetin to an artificial diet, its role in defense mechanisms against phytophagous insects was studied. We try to lay a preliminary basis to explore the mechanisms used by the fall webworm to adapt to quercetin present in host plants.

## 2. Materials and Methods

### 2.1. Insect

The *Hyphantria cunea* eggs were sterilized using 10% formaldehyde solution for 15 min and the larvae hatched from these eggs were reared on an artificial diet. The insects were reared and maintained in a climate chamber (RXZ-500B, Jiangnan Instrument Factory, Foshan, China) at $25 \pm 1$ °C, $70 \pm 5$% relative humidity, and 16:8 h (L:D) photoperiod, according to Cao [41]. The larval molting times were recorded. The third-instar *H. cunea* was transferred to a quercetin-treated diet at various doses. Third- or fifth-instar larvae were randomly selected for the follow-up experiments.

### 2.2. Feeding Treatment

Quercetin was purchased from Shanghai Yuan Ye Bio-Technology Co., Ltd., Shanghai, China. The impact of quercetin on growth and development, mortality, nutritional indices, and the activity or content of detoxification enzymes was determined after adding different concentrations of quercetin to an artificial diet [41]. Briefly, quercetin was weighed, dissolved in 20% dimethyl sulfoxide (DMSO), and serially diluted to gradient concentrations



of 0.25, 0.50, 1.00, 2.00, and 4.00% (*w/v*), respectively, based on the content of quercetin present in the leaves of several host plants of *H. cunea* [42,43]. The solution was evenly mixed with the artificial diet and constantly stirred. The diet was then poured into rearing cups (200 mL each) and allowed to solidify. Each cup was filled with 15 freshly molted third- or fifth-instar *H. cunea* larvae. All rearing cups were placed in an incubator. The larvae of the control group were fed a normal diet to which the same volume of 20% DMSO was added without quercetin.

### 2.3. Growth and Development Assay

Third-instar larvae with the same developmental stage was selected as the experimental insects. After 12 h starvation, the larvae were exposed to the artificial diet containing 0.50% quercetin. A normal artificial diet with 20% DMSO was used as the control. Each concentration had three replicates, and each cup included thirty larvae. The method was based on Pan [44]. The ecdysis duration, larval survival rate, pupation rate, emergence rate, sex ratio, and number of eggs laid by each female after mating were recorded.

### 2.4. Mortality Assay

Newly molted fifth-instar larvae were fed on diets containing 0.25, 0.50, 1.00, 2.00, and 4.00% quercetin for six days. Larvae fed artificial diets supplemented with 20% DMSO were used as the control. Feeding conditions were the same as described in Section 2.1. Larvae were observed daily to evaluate mortality. The number of dead larvae was recorded at the same time every day. Each replicate included ten larvae and all the trials were repeated four times. Determination of mortality was done according to Pan [44]. Larval mortality (%) was calculated as follows: (number of dead larvae/total number of experimental larvae) × 100%.

### 2.5. Nutritional Effect Assay

The fifth-instar *H. cunea* was used to study the nutritional effects of quercetin. After starvation for 24 h, fifteen larvae were fed with a quercetin diet for 48 h. Feces from the larvae were collected after the removal of the diet from the larvae after 12 h. Each treatment was thrice repeated. The decrease in mass from pre-feeding was assessed after the feeding experiments by weighing the larvae, excreted waste, and residual feed after being dried at 80 °C for 8 h to achieve a consistent weight. Additional replicates were simultaneously employed to measure both the fresh and dried weights. The dry weights of the pre-feeding larvae and diet were calculated based on the larval water content.

Approximate digestibility (AD), efficiency of conversion of digested food to body substance (ECD), efficiency of conversion of ingested food to body substance (ECI), relative growth rate (GR), and consumption index (CI) were selected as nutritional indices [45], and dry weight was used to determine them. Efficiency of conversion of ingested food (ECI) and digested food (ECD) refer to the mass added per unit of food consumed and absorbed by the insect, respectively [46].

CI was calculated as follows:

CI = F/T·A

F = fresh or dry weight of the consumed food

T = duration of the feeding period

A = mean fresh or dry weight of the larvae during the feeding period (as used in the calculation of GR below)

Relative GR was calculated as follows:

GR = G/T·A

G = fresh or dry weight gain of the larvae during the feeding period

ECI was calculated as follows:

ECI = (wt gained/wt of food ingested) × 100

wt = weight (as used in the calculation of AD and ECD below)

AD was calculated as follows:

AD = [(wt of food ingested − wt of feces)/wt of food ingested] × 100
ECD was calculated as follows:
ECD = [wt gained/(wt of food ingested − wt of feces)] × 100

### 2.6. Detoxification Enzyme Activity or Concentration Assays

Detoxification enzymes of the fifth-instar larvae fed on diets containing different quercetin concentrations i.e., 0.25, 0.50, 1.00, 2.00, and 4.00% [*w/v*] for 36 h were determined. Each treatment group was thrice repeated using five larvae in each. After cleansing in ice-cold physiological saline, midgut tissues of each treated larvae were dried using filter paper. The samples were then weighed and placed in a polyethylene tube. A volume of cold phosphate buffer (pH 7.0) nine times the weight of the midgut of the larvae was added. The tissue sample was pulverized at 10,000–15,000 rpm to create a tissue homogenate, which was then centrifuged at 8000 rpm for 10 min at 4 °C. The supernatant was collected to measure enzyme activity or concentration. The procedure for each enzyme activity or concentration assay was performed according to the manufacturer's instructions for each kit. The GST assay kit (product ID: YX-W-A204) was purchased from Shanghai You Xuan Biotechnology Co., Ltd., Shanghai, China. The CarE assay kit (product ID: ml036265), UGT assay kit (product ID: ml062849), and ABC transporter assay kit (product ID: ml74) were purchased from Shanghai Enzyme-Linked Biotechnology Co., Ltd., Shanghai, China. The insect CYP450 ELISA k3416it (product ID: JL22832) was purchased from Shanghai Jianglai Industrial Limited by Share Ltd., Shanghai, China. The method used was based on Pan [44]. Total protein content in each enzyme stock solution was measured using a protein quantitative assay kit (product ID: PW0103) from Biomiga.

### 2.7. Statistical Analysis

All results are expressed as the mean ± SE. Mortality, nutritional indices, and detoxification enzyme activity or concentration were analyzed using a standard one-way analysis of variance (ANOVA, Turkey's test, $p < 0.05$), implemented in SPSS 26.0. Student's *t*-test at $p < 0.05$ was used to compare the growth and development of *H. cunea* between 0.50% quercetin concentration and control (20% DMSO). Origin 8.0 software was used to draw figures.

## 3. Results

### 3.1. Effects of Quercetin on the Growth and Development of H. cunea

As shown in Table 1, the development time from the third-instar to the pupation stage was considerably longer in the treatment groups (0.50%) compared with the control group, with a difference of 4.4 days (t = −24.489; df = 4; $p = 0.0001$). In particular, the total survival rate of larvae (t = 8.819; df = 4; $p = 0.001$), pupation rate (t = 4.839; df = 4; $p = 0.008$), and sex ratio (t = 12.490; df = 4; $p < 0.0001$) were significantly decreased (Table 2). However, no significant effects were observed on emergence rate (t = −1.212; df = 4; $p = 0.292$) or fecundity (t = 1.422; df = 4; $p = 0.167$) (Table 2).

**Table 1.** The effect of 0.50% quercetin on the developmental duration of third-instar *Hyphantria cunea* larvae.

| Treatment | Developmental Duration of Larval Instar (Day) | | | | | Larval Period from the 3rd Instar (Day) |
|---|---|---|---|---|---|---|
| | 3rd Instar | 4th Instar | 5th Instar | 6th Instar | 7th Instar | |
| Control | 3.0 ± 0.00 b | 5.4 ± 0.09 b | 4.7 ± 0.05 b | 3.8 ± 0.07 a | 4.1 ± 0.11 b | 21.2 ± 0.13 b |
| Quercetin | 4.2 ± 0.06 a | 7.1 ± 0.09 a | 5.8 ± 0.09 a | 3.8 ± 0.12 a | 4.4 ± 0.11 a | 25.6 ± 0.11 a |

The data presented in this table is mean ± SE. Different letters in the column show a significant difference (Student's *t*-test, $p < 0.05$). Control, artificial diet containing 20% DMSO.

**Table 2.** The effects of 0.50% quercetin on developmental parameters of third-instar *Hyphantria cunea*.

| Treatment | Total Survival Rate of Larvae (%) | Pupation Rate (%) | Emergence Rate (%) | Sex Ratio (m/f) | Number of Eggs Laid Per Female |
|---|---|---|---|---|---|
| Control | 74.44 ± 1.14 a | 100 ± 0.00 a | 92.54 ± 1.17 a | 0.72 ± 0.017 a | 673.39 ± 34.80 a |
| Quercetin | 60.00 ± 1.18 b | 94.44 ± 1.15 b | 94.12 ± 1.13 a | 0.46 ± 0.023 b | 592.80 ± 43.03 a |

The data presented in this table is mean ± SE. Different letters in the column show a significant difference (Student's *t*-test, $p < 0.05$). M, male; f, female. Control, artificial diet containing 20% DMSO.

### 3.2. Effects of Quercetin on the Mortality of H. cunea

The fifth-instar *H. cunea* that were fed with different concentrations of quercetin began to die after three days. A positive correlation was observed between mortality and the concentration of quercetin (Figure 1). On day 4, quercetin treatment (≥1.00%) significantly increased *H. cunea* mortality (F = 7.150; df = 5, 18; $p = 0.001$) compared with controls. No significant differences were observed in the mortality of *H. cunea* on day 5 (F = 32.600; df = 5, 18; $p < 0.0001$) and 6 (F = 38.415; df = 5, 18; $p < 0.0001$) between different quercetin concentrations.

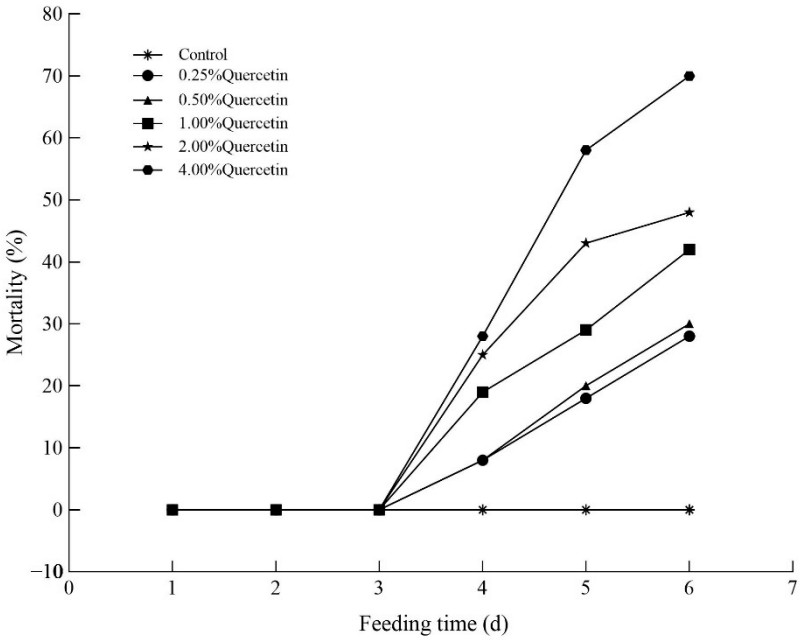

**Figure 1.** The mortality of fifth-instar *Hyphantria cunea* fed on diets containing different concentrations of quercetin. Significant differences among treatments were calculated using Tukey's test at $p < 0.05$. Control, artificial diet containing 20% DMSO.

### 3.3. Effects of Quercetin on Nutrition in H. cunea Larvae

Results showed that quercetin significantly affected AD (F = 67.991; df = 5, 12; $p < 0.0001$) and CI (F = 83.718; df = 5, 12; $p < 0.0001$) of *H. cunea* larvae and showed a dose-dependent decrease with increasing concentrations, reaching a minimum at 4% (Table 3). As observed in Table 3, ECI (F = 5.476; df = 5, 12; $p = 0.007$) and GR (F = 8.749; df = 5, 12; $p = 0.001$) significantly decreased at only 4%, whereas the rest showed no significant change compared with controls. However, ECD (F = 2.320; df = 5, 12; $p = 0.108$) showed no discernible changes.

**Table 3.** Nutritional indices of fifth-instar *Hyphantria cunea* fed on quercetin and control diets.

| Quercetin Concentration (%) | AD (%) | ECD (%) | ECI (%) | GR (g/g·d) | CI (g/g·d) |
|---|---|---|---|---|---|
| Control | 0.885 ± 0.001 a | 0.113 ± 0.013 a | 0.100 ± 0.010 a | 0.200 ± 0.011 a | 1.992 ± 0.068 a |
| 0.25 | 0.848 ± 0.035 ab | 0.140 ± 0.004 a | 0.120 ± 0.006 ab | 0.207 ± 0.008 a | 1.721 ± 0.028 b |
| 0.50 | 0.757 ± 0.019 b | 0.227 ± 0.001 a | 0.172 ± 0.010 a | 0.260 ± 0.012 a | 1.513 ± 0.040 c |
| 1.00 | 0.646 ± 0.016 bc | 0.270 ± 0.031 a | 0.174 ± 0.017 a | 0.244 ± 0.025 a | 1.401 ± 0.014 cd |
| 2.00 | 0.602 ± 0.024 c | 0.301 ± 0.076 a | 0.182 ± 0.042 a | 0.235 ± 0.049 a | 1.300 ± 0.031 d |
| 4.00 | 0.326 ± 0.035 d | 0.183 ± 0.086 a | 0.054 ± 0.020 b | 0.051 ± 0.020 b | 0.912 ± 0.040 e |

The data presented in this table is mean ± SE. Different letters in the column show a significant difference (Tukey's test, $p < 0.05$). g (gram); d (day). Control, artificial diet containing 20% DMSO.

*3.4. Effects of Quercetin on the Detoxification Enzymes in H. cunea Larvae*

Activities or contents of the five detoxification enzymes in the midgut of fifth-instar *H. cunea* were determined, following treatment with different quercetin concentrations (0.25, 0.50, 1.00, 2.00, and 4.00%) for 36 h. Activities or concentrations of GSTs (F = 5.694; df = 5, 12; $p < 0.001$), UGT (F = 12.890; df = 5, 12; $p = 0.004$), ABC transporters (F = 7.240; df = 5, 12; $p < 0.001$), and CarE (F = 3.578; df = 5, 12; $p = 0.023$) in the fifth-instar larvae of *H. cunea* were significantly altered depending on quercetin concentration in the diet. In particular, the activity of UGT (Figure 2D) and concentration of ABC transporters (Figure 2C) responded to quercetin in a concentration-dependent manner, but the difference between quercetin and control groups was only significant when the concentration of quercetin was higher than 0.50%. The activity of GSTs significantly increased with an increase in quercetin concentration above 1.00% (Figure 2B). Conversely, the concentration of CarE was reduced by a quercetin concentration of 0.50%–2.00% (Figure 2E). Quercetin did not significantly alter CYP450 (F = 7.240; df = 5, 12; $p = 0.970$; Figure 2A) activity at any concentration tested (Figure 2A). The present study found that different concentrations of quercetin induced differential activities or contents in the detoxification enzymes in the fifth-instar larvae of *H. cunea*, thereby demonstrating that different types of detoxification enzymes were activated at different stages.

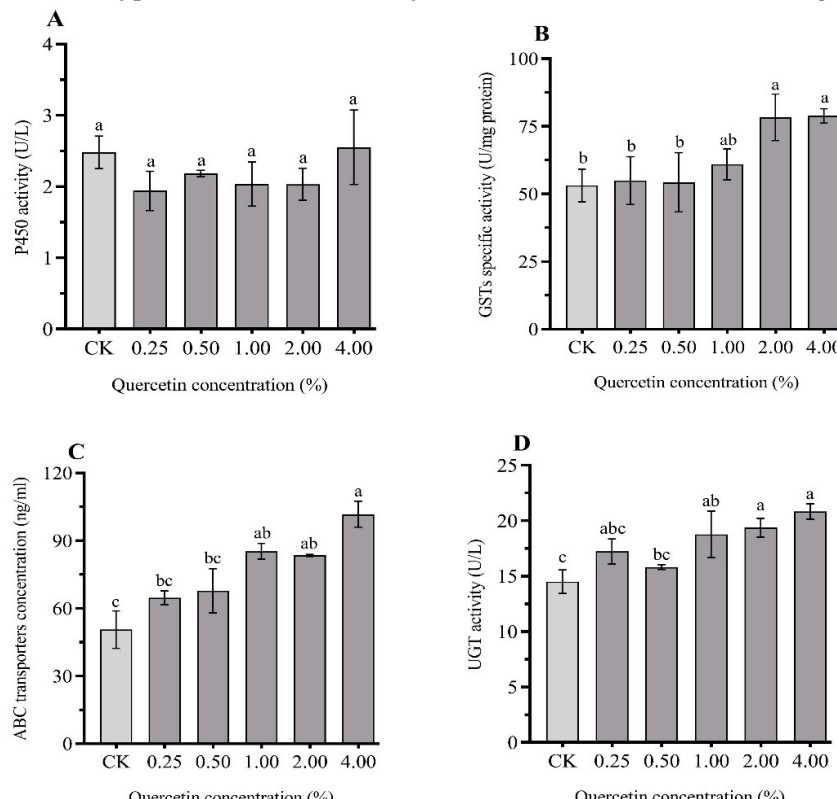

**Figure 2.** *Cont.*

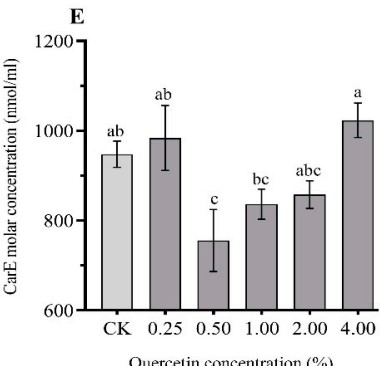

**Figure 2.** The activity or concentration of five detoxification enzymes (P450, GSTs, ABC transporters, UGT, and CarE) in fifth-instar larvae of *Hyphantria cunea* fed on different concentrations of quercetin for 36 h. The enzymatic activity or concentration of P450 (**A**), GSTs (**B**), ABC transporters (**C**), UGT (**D**), and CarE (**E**) was measured for different quercetin concentrations. Different letters above the bars show significant differences at $p < 0.05$ (Tukey's test). CK: artificial diet containing 20% DMSO without quercetin. CYP450 = P450.

## 4. Discussion

Secondary metabolites of host plants cause toxicity to insects by interfering with their essential metabolic, biochemical, and physiological functions [47]. At present, a large number of toxicological studies reported adverse effects of secondary metabolites on the growth and development of various insects at the biochemical and molecular level These studies also emphasized variations in toxic effects on insects with different secondary metabolite compounds and concentrations [48,49]. Plant secondary metabolites and other attractants are poisonous, and repellent to pollinators that try to eat flowers [48]. In living plants, quercetin can operate as a secondary metabolite to deter insect herbivory, with Hemiptera, Diptera, and Lepidoptera being much more susceptible to its toxicity [50]. As a successful invasive pest, the fall webworm rapidly spread in China by virtue of its robust adaptability to a wide range of hosts. In the current study, the effect of quercetin on the growth and development, mortality, nutritional index, and activity or content of detoxification enzymes of *H. cunea* was investigated.

Quercetin had a significant impact on the growth and development of *H. cunea* (Tables 1 and 2). This was in accordance with Gikonyo [51], who showed that quercetin inhibited growth and had a toxic impact on *Aedes aegypti* L. (Diptera: Culicidea). Quercetin was also reported to decrease the total development period of *Bactrocera cucurbitae* Coquillett (Diptera: Tephritidae) [52] and the larval duration of *Drosophila melanogaster* Meigen (Diptera: Drosophilidea) at a concentration of 1.75% [13]. These results indicated that the effect of quercetin varied between insect species in a concentration-dependent manner. Meanwhile, in Lepidoptera, Shi [53] showed that quercetin might impair the immune system while also inhibiting growth and development, leading to an increase in the death rate in silkworm (Lepidoptera: Bombycidae). In the present study, low concentrations of quercetin (0.50%) extended the larval development period and inhibited their growth and development through cumulative toxic effects. Selin-Rani [31] found that the mean larvae weight and fecundity of *S. litura* were reduced when fed a quercetin diet (2, 4, or 6 ppm).

The death of *H. cunea* larvae on the fourth day of quercetin treatment (Figure 1) suggested that quercetin inhibited growth and development through cumulative toxic effects, caused by the accumulation of quercetin in *H. cunea* larvae. Mallikarjuna [54] showed that the interspecific derivatives of groundnut, containing high levels of quercetin, induced high mortality in *S. litura*. In addition, our analyses of the larval midguts also showed that mortality increased with an increase in quercetin concentration (Figure 1), which was consistent with the results of Ateyyat [55] that reported that quercetin reduced the survival rate of *Eriosoma lanigerum* (Homoptera: Pemphigidea). Cui [33] and Wang [21] demonstrated that the mortality of *O. asiaticus* larvae displayed a positive relationship with

the amount of quercetin added to their diet. These results suggest that quercetin may be an effective botanical insecticide for polyphagous pests, such as *H. cunea*.

Earlier reports indicated that nutritional indices directly reflected the food use of herbivorous insects [56]. Here, 0.25%–4.00% of quercetin reduced AD and CI, 0.25%–2.00% increased ECD, ECI, and GR, and 4% decreased ECI and GR. At quercetin concentrations of 0.25%–2.00%, a decrease in AD was compensated by an increase in ECD with no significant change in ECI and GR (Table 3). Studies showed that quercetin content was low in most host plants of *H. cunea*, including *Morus alba* L. (Urticales: Moraceae) (0.18%–1.48%) and *Lonicera japonica* Thunb. (Dipsacales: Caprifoliaceae) (11%–0.13%) [42,57,58]. Hence, by adjusting its nutritional strategies, *H. cunea* adapted to the presence of quercetin in its host plants. The findings depicted in Table 3 imply that nutritional indicators exhibit a dose-response effect. Except for the highest quercetin concentration, values for AD and CI decreased, whereas ECD and ECI values increased with increasing quercetin concentrations. However, values for GR were subject to fluctuation. It was also demonstrated that quercetin stimulated nutritional indices most effectively in *H. cunea* larvae at 4% quercetin. We did not investigate other plant secondary metabolites, but a similar study revealed that gallic acid also affected food conversion efficiency in *H. cunea*. In this study, quercetin decreased AD and raised the value of ECI. Our current results were consistent with the nourishing effect of gallic acid on *H. cunea*, as observed by Wu [59], which showed that gallic acid-reduced AD in *H. cunea* may lower the toxicity of phenolic secondary metabolites by reducing food intake and by raising ECI to satisfy their nutritional needs for growth and development. We speculated that *H. cunea* may fight against the toxicity of quercetin by decreasing the efficiency of food consumption and enhancing the efficiency of food utilization. This resistance to botanical toxins by insects may also have occurred at the expense of increased energetic costs [10,60]. Higher concentrations of quercetin in the body of *H. cunea* would result in high energy consumption to mediate detoxification; the converted energy would not maintain the normal growth of larvae, which explains why higher concentrations of quercetin significantly inhibited growth and resulted in high mortality of *H. cunea*.

We demonstrated that quercetin increased the activity of GSTs (Figure 2B), UGTs (Figure 2D), and the concentration of ABC transporters (Figure 2C), indicating that these enzymes play a key role in the regulation of detoxification enzymes in *H. cunea*. Large multigene families of ABC transporters present in insects were involved in the efflux of toxic/unwanted compounds derived from diet, endogenous metabolism, and chemical pesticides [61]. GSTs are ubiquitous enzymes that catalyze the conjugation of reduced glutathione (GSH) with xenobiotic substances, resulting in lower toxicity and increased solubility of harmful chemicals for excretion [62–64]. Li [65] showed that GSTs in *S. litura* were active either metabolically or as an antioxidant system to resist fenvalerate and cyhalothrin. In this study, high doses (2 and 4%) of quercetin induced the maximum activity of GSTs (Figure 2B), proving that GSTs were more sensitive to quercetin at higher concentrations. Additionally, *H. cunea* larvae fed on 0.50%–2.00% of quercetin showed a decrease in CarE concentration (Figure 2E) compared with the control group. A similar result was reported by Wang [34], where reduced CarE activity induced by quercetin treatment was observed in the larvae of *L. dispar*. Conversely, Chen [39] found that 0.10% quercetin increased the activity of CarE in *H. armigera*. Together, these results suggest that changes in CarE activity in response to quercetin differ between insect species. Besides, the CYP450 activity of *Bombyx mori* L. (Lepidoptera: Bombycidae), *H. armigera*, and *S. exigua* was found to increase after being fed a quercetin diet [29,30,66]. Chen [39] found that the activity of CYP450 and the relative expression levels of three CYP450 genes increased in *H. armigera* after exposure to quercetin for 48 h, which contributed to an increase in lambda-cyhalothrin insensitivity in *H. armigera* larvae. However, in our study, quercetin did not change the activity of CYP450. It was also speculated that interactions may occur between the detoxification enzyme genes, which necessitates further research to determine the specific functions of each detoxification enzyme gene.

In-depth molecular studies about the underlying mechanisms of detoxification enzymes are required to comprehend the physiological adaptation of *H. cunea* against host plant secondary metabolites. Ingestion of plant secondary metabolites by insects activates an internal detoxification mechanism, involving changes in gene expression or regulation of detoxification enzyme activity [9,67].

## 5. Conclusions

The current study provides evidence of multiple detoxification enzymes involved in the metabolism of quercetin in *H. cunea* larvae. We conclude that quercetin delayed the larval growth and development of *H. cunea* by altering nutrient use and detoxification enzyme activities or contents. These results provided key information about the underlying adaptation mechanisms of *H. cunea* to quercetin. However, further studies are needed to investigate the expression patterns of associated genes and regulatory mechanisms of *H. cunea* in response to quercetin. These findings may be useful for developing environmentally friendly pesticides for controlling this key pest.

**Author Contributions:** Methodology, X.M.; investigation, Z.-Y.P. and Y.-F.Y.; resources, H.-Y.L.; writing—original draft preparation, X.M. and Y.-L.G.; writing—review and editing, Y.-L.G.; supervision, M.C.; funding acquisition, M.C. All authors have read and agreed to the published version of the manuscript.

**Funding:** This research was supported by China's National Key R&D Program (2021YFD1400300).

**Data Availability Statement:** Not applicable.

**Conflicts of Interest:** The authors declare no conflict of interest.

## Abbreviations

Approximate digestibility, AD; efficiency of conversion of digested food to body substance, ECD; efficiency of conversion of ingested food to body substance, ECI; relative growth rate, GR; consumption index, CI; cytochrome P450 monooxygenases, CYP450; glutathione S-transferases, GSTs; carboxylesterases, CarE; UDP-glycosyltransferases, UGTs; ATP-binding cassette transporters, ABC transporters.

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
