# Peer review of "The Effect of Quercetin on the Growth, Development, Nutrition Utilization, and Detoxification Enzymes in Hyphantria cunea Drury (Lepidoptera: Arctiidae)"

_forests, doi:10.3390/f13111945_

Round 1
Reviewer 1 Report
The article brings some new data on the effects of a plant secondary metabolite on a quarantine pest. It is well-written and gives more information on the adaptation mechanisms of H. cunea to quercetin. The only concern is the lack of diet control (without DMSO) and it needs a better explanation of the statistical methods. I would suggest the alteration of the title to - the effect of quercetin on the growth, development, nutrition utilization, and detoxification enzymes in Hyphantria cunea Drury (Lepidoptera: Arctiidae), but leave to the author to decide.
As an overall comment to the introduction section, I find it clearly written, but identified a missing link, could you explain if quercetin is present in plants attacked by the pest, or in plants not attacked by it (or both); it is stated in the methods and discussion but it makes sense here in the introduction too. The explanation of the reason for choosing quercetin (I believe it was because of its effect on other lepidopterans and insects) should be clearly stated here.
Some small things could be corrected as I suggest:
L.15 - add a space between the number and %
L.35 - add a space before [7]
L.42 - please clarify “and render them less toxic and easier to transport or excrete”
L.62 - misses something before “assumed”, maybe “its role is assumed to be the protection”
L.61-71 - when a species is referred to for the first time please add the descriptors name too
L.64 - (Lepidoptera: Noctuidae) is repeated here, please eliminate it
L.64 - I suggest the simplification of this phrase: “Quercetin inhibits the growth and reduces the pupation rate of Helicoverpa armigera (Lepidoptera: Noctuidae) and Spodoptera exigua (Lepidoptera: Noctuidae) [25-28], and increases the larval mortality of S. frugiperda, Oedaleus asiaticus (Orthoptera: Acrididae), and Lymantria dispar (Lepidoptera: Lymantriidae) [29-31].”
Regarding the methods section, please add a space between the number and its unity of measure.
Regarding feeding treatment, it means that 20 % of the artificial diet was replaced by DMSO or DMSO+quercetin? Please clarify. Firstly, it is stated that quercetin is dissolved in 20 % DMSO and then the replacement of quercetin by DMSO is not clear.
L.102 - what do you mean by “similar status”, I assume that if they are 3rd instar larvae they have the same developmental stage, or is it related to another characteristic?
L.104 - did you previously evaluated, or was studied in other works, the possible effect of DMSO addition to the artificial diet of H. cunea? This would be necessary to validate the results
L.124 - replace “computed” with calculated
In the 2.5. section I suggest the separation of the different formulas, as it is difficult to understand in the present state
L.149 - each treatment or each treated larvae?
In 2.7. section, please add the software you used for ANOVA’s, and explain why you did t-test instead of a posthoc test after the ANOVA. Why do you only compare one of the concentrations with the control regarding t-test?
L.179 - the table caption should explain all its content, and the nomenclature used in the text should be maintained in the table.
L.187 - replace 3 with three here
L.195 - here you used the Tukey posthoc test, but it is not referred to in the methods, please add this there
L.218 - please replace greater with higher
L. 225 please replace “may be” with might exist; this last phrase could be placed in the discussion
L.237 - please add some examples (references) at the end of the sentence.
L.241 - there is more spacing than it should be at the beginning of the sentence
L.246-248 - please add the species descriptors; these are all dipterans, care should be taken to compare these results with lepidopterans
L.256 - inhibited the larval growth and seems to have cumulative toxic effects? Or inhibited the larval growth through cumulative toxic effects?
L.259 - you shall not make direct comparisons between insects from different orders, I believe, at least state here which insect you are comparing your results to.
L.263 - same here
L.272-274 - it makes sense, but please link this with the previous findings regarding nutritional indices, as it is not clear
L.298 - Bombix mori is first referred to here, so please add the complete species name (not abbreviated), the descriptor, and order:family.
L.313 - “and that H. cunea activates physiological and chemical defense mechanisms in host plants” In this work, this was not observed. Is this based on other references? If so please add them.
Author Response
Dear Editor and Reviewer,
We quite appreciate your favorite consideration and the reviewer’s insightful comments concerning our manuscript entitled “The effect of quercetin on the growth, development, nutrition utilization, and detoxification enzymes in Hyphantria cunea Drury (Lepidoptera: Arctiidae)” (ID: 1991642). Those comments are very valuable and helpful for improving the quality and readability of our paper, as well as the important guiding significance to our future research. We have studied the comments carefully and have revised the paper exactly according to the reviewer’s comments. We hope this revision can meet with approval. The revised portions are marked in red in the paper and the main revisions corresponding to the reviewer’s comments are as follows.
Please see the attachment.

Reviewer 2 Report
1. In L125-126 ECD and ECI have identical definitions but differ in the estimation formula. What exactly is their difference?
2. Almost all nutritional indices exhibit a consistent pattern of increase/decrease except for the highest –i.e. 4%-quercetin concentration. Moreover, are all values in Table 3 percentages or ratios? Their definition is not always expressed in percentages, i.e. CI = F / T∙A. They rather seem to be ratios. Also, between T and A the symbol ‘ ∙ ‘ should be inserted.
3. References 45-47 showed that quercetin inhibited growth in dipteran, not in Lepidoptera where H. cunea belongs.
4. The findings of this study to be in accordance with that of Wu (ref 53) should be discussed in terms of the concentration of quercetin. The findings depicted in Table 3 imply a dose-response effect, which is not discussed at all.
5. Lines 255-257 oppose lines 2277-279. In addition, the MS does not examine other secondary metabolites and for this, it is not adequate to state that quercetin increases food conversion efficiency.
6. In Fig. 2 there are no consistent y-axes. It would facilitate the general reader to have the same units in the y-axis, preferably activity units (U/L).
7. In addition, in Fig. 2 more enzymes show the same or similar dose-response curve but the authors restrict their discussion to enzymes GST, ABC, UGT.
8. In Fig. 2, is it possible to have a ‘bc’ or ‘abc’ significance of bars without having a ‘b’ significance level?
9. In Fig. 2, what is CK? The legend does not explain this. Also, Cytochrome P [=CYP450] or similarly P450.
10. Unavoidably, all similar MS is full of abbreviations for enzyme names, measures of food efficiency, etc. For this, I suggest existing a list of abbreviations at the beginning of the MS in order to facilitate the general reader.
Author Response

(The authors gave the same response as above.)

Reviewer 3 Report
Manuscript-ID: forests-1991642 Title of the study: Quercetin affects the growth, development, nutrition utilization, and induces detoxification enzymes in Hyphantria cunea Drury (Lepidoptera: Arctiidae)
In this study, Yi-Lin Gao et al. showed that different physiological mechanisms can be induced to avoid the harmful effects of Quercetin in Hyphantria cunea Drury (Lepidoptera: Arctiidae). Quercetin was found to have a significant effect on food utilization by H. cunea larvae. The larvae responded to Quercetin stress by increasing the activity of detoxification enzymes, namely, glutathione S-transferases, UDP-glucuronosyl-transferases, and ATP-binding cassette transporters, while, quercetin inhibited the activity of carboxylesterases. These findings are of a huge importance to determine the mode of adaptation of H. cunea against secondary metabolites (herein, Quercetin) in the host plant. Although the paper is original, and the importance of these findings, I recommend revising the paper following the comments below.
The methodology adopted in this study still raises some concerns as the authors did not cite any relevant reference for the sections :
2.3. Growth and Development Assay
2.4. Mortality Assay
2.5. Nutritional Effect Assay
2.6. Detoxification Enzyme Activity Assays
Author Response

(The authors gave the same response as above.)
